# Functional partitioning of a liquid-like organelle during assembly of axonemal dyneins

Chanjae Lee[1], Rachael M Cox[1], Ophelia Papoulas[1], Amjad Horani[2], Kevin Drew[1], Caitlin C Devitt[1], Steven L Brody[3], Edward M Marcotte[1]\*, John B Wallingford[1]\*

[1]Department of Molecular Biosciences, University of Texas, Austin, United States; [2]Department of Pediatrics, Washington University School of Medicine, St. Louis, United States; [3]Department of Medicine, Washington University School of Medicine, St. Louis, United States

**Abstract** Ciliary motility is driven by axonemal dyneins that are assembled in the cytoplasm before deployment to cilia. Motile ciliopathy can result from defects in the dyneins themselves or from defects in factors required for their cytoplasmic pre-assembly. Recent work demonstrates that axonemal dyneins, their specific assembly factors, and broadly-acting chaperones are concentrated in liquid-like organelles in the cytoplasm called DynAPs (Dynein Axonemal Particles). Here, we use *in vivo* imaging in *Xenopus* to show that inner dynein arm (IDA) and outer dynein arm (ODA) subunits are partitioned into non-overlapping sub-regions within DynAPs. Using affinity- purification mass-spectrometry of in vivo interaction partners, we also identify novel partners for inner and outer dynein arms. Among these, we identify C16orf71/Daap1 as a novel axonemal dynein regulator. Daap1 interacts with ODA subunits, localizes specifically to the cytoplasm, is enriched in DynAPs, and is required for the deployment of ODAs to axonemes. Our work reveals a new complexity in the structure and function of a cell-type specific liquid-like organelle that is directly relevant to human genetic disease.

**\*For correspondence:**
marcotte@icmb.utexas.edu
(EMM);
wallingford@austin.utexas.edu
(JBW)

**Competing interests:** The authors declare that no competing interests exist.

## Introduction

Motile cilia are microtubule-based cellular projections and their oriented beating generates fluid flows that are critical for development and homeostasis. Ciliary beating is driven by a complex set of axoneme-specific dynein motors that drive sliding of the axonemal microtubule doublets. Based on their relative positions within the axoneme, these motors are classified as either outer dynein arms (ODAs) or inner dynein arms (IDAs)(*Figure 1A*, upper inset), the former driving ciliary beating generally, and the latter tuning the waveform (*Kamiya and Yagi, 2014*; *King, 2018*). Mutations in genes encoding ODA or IDA subunits are the major cause of the motile ciliopathy syndrome known as primary ciliary dyskinesia (PCD; MIM 244400). This genetic disease results in repeated sinopulmonary disease, bronchiectasis, cardiac defects, situs anomalies, and infertility (*Horani et al., 2016*; *Mitchison and Valente, 2017*; *Wallmeier et al., 2020*). Interestingly, PCD can also result from mutations in genes encoding any of several cytoplasmic proteins collectively known as Dynein Axonemal Assembly Factors (DNAAFs)(*Desai et al., 2018*; *Fabczak and Osinka, 2019*).

Axonemal dynein motors were known to be pre-assembled in the cytoplasm before deployment to cilia (*Fowkes and Mitchell, 1998*), but the first description of DNAAFs came later, with the identification of *KTU* (aka *DNAAF2*)(*Omran et al., 2008*). Studies of motile ciliopathy patients have now defined an array of cytoplasmic DNAAFs that are never part of the axoneme, yet when mutated result in loss of axonemal dyneins, and in turn, defective cilia beating (*Diggle et al., 2014*; *Horani et al., 2012*; *Horani et al., 2013*; *Kott et al., 2012*; *Mitchison et al., 2012*; *Moore et al.,*

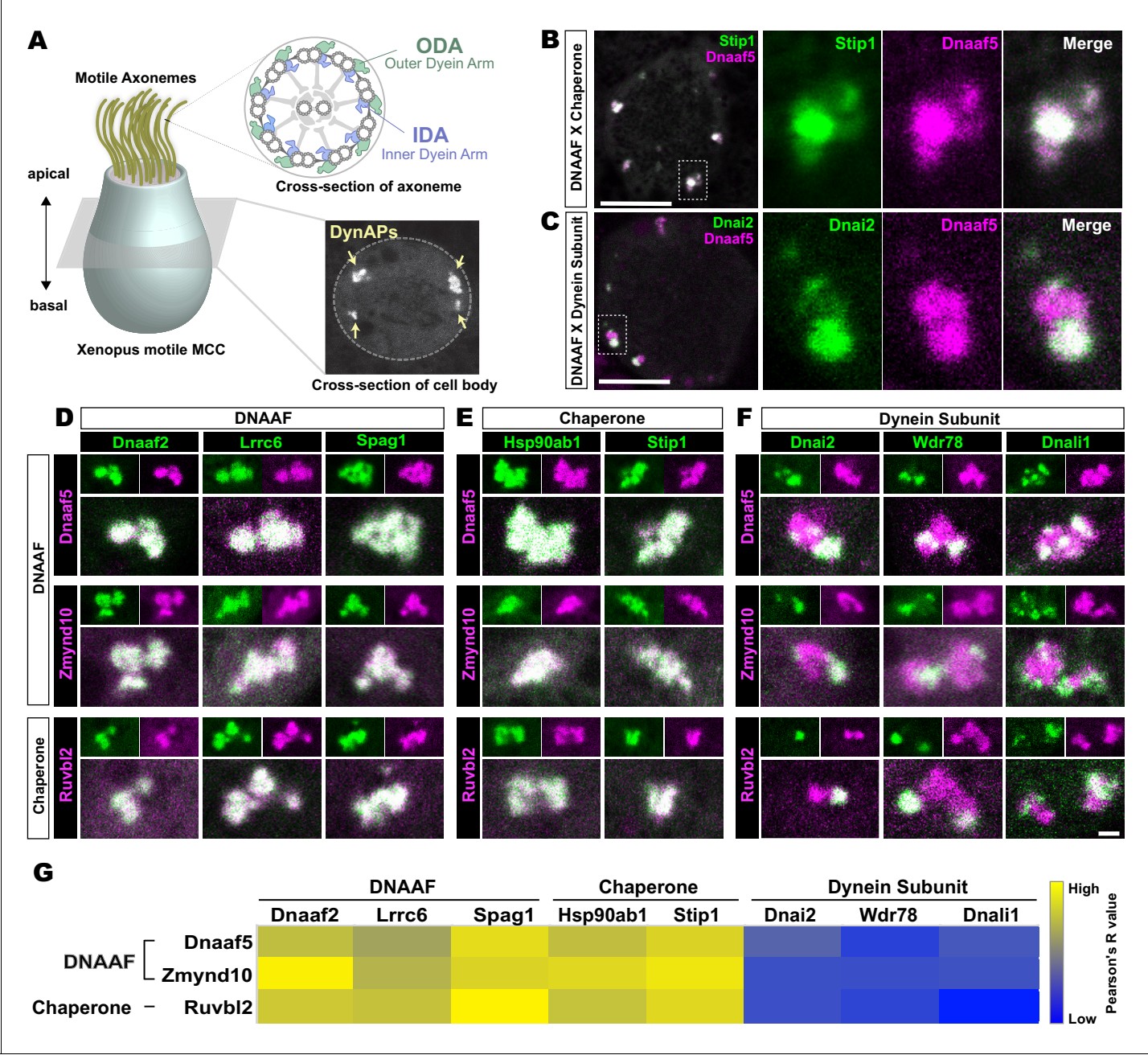

**Figure 1.** Dynein subunits occupy sub-regions within DynAPs. (A) Schematic showing an MCC. The upper inset shows a schematic cross section through an axoneme indicating the relative positions of ODAs and IDAs; the lower inset shows a representative *en face* optical section through the MCC cytoplasm with dynAPs indicated DynAPs. (B) *En face optical* section through the cytoplasm of a *Xenopus* MCC expressing GFP-Stip1 (a chaperone, green) and mCherry-Dnaaf5 (a DNAAF, magenta). Note near-perfect co-localization. Dashed box indicates region shown in high magnification, split-channel views at right. Scale bars = 10 μm. (C) GFP-Dnai2 (a dynein subunit, green) co-localizes only partially with mCherry-Dnaaf5. (D, E) Images showing enlarged views of individual DynAPs after pairwise labeling with the indicated tagged DNAAFs or chaperones. Smaller panels show split channels, larger images show the merged channels. Note high degree of co-localization. (F) Similar pair-wise labeling with dynein subunits reveals only partial co-localization. Scale bar = 1 μm for panels D-F. (G) Heatmap showing Pearson correlation for colocalization of GFP fusion proteins with mCherry fusion proteins. Yellow indicates high value (Pearson's R value = 0.85), while blue indicates low value (Pearson's R value = 0.45). Exact Pearson's R values for each protein combination can be found in *Figure 1—figure supplement 1*.

The online version of this article includes the following figure supplement(s) for figure 1:

**Figure supplement 1.** Quantification of localization data.

*2013*; *UK10K Rare Group et al., 2017*; *Paff et al., 2017*; *UK10K et al., 2013*; *Zariwala et al., 2013*). These specialized DNAAFs are now known to act in concert with ubiquitous, broadly-acting chaperones of the Heat Shock Protein (Hsp) Family (*Cho et al., 2018*; *Li et al., 2017*; *Mali et al., 2018*; *UK10K Rare Group et al., 2017*; *Omran et al., 2008*; *Paff et al., 2017*; *UK10K et al., 2013*; *Yamaguchi et al., 2018*; *Yamamoto et al., 2010*; *Zhao et al., 2013*; *Zur Lage et al., 2018*).

How DNAAFs organize the assembly of axonemal dyneins is a complex question, as each axoneme contains an series of closely related multiprotein complexes (*King, 2018*). For example, ODAs and IDAs incorporate diverse heavy, intermediate, and light chains encoded by distinct genes; few subunits are shared. Moreover, at least eight distinct IDA sub-types and two ODA sub-types have been described (*Dougherty et al., 2016*; *Fliegauf et al., 2005*; *King, 2018*). Generally, disruption of any single DNAAF has been found to impact both ODAs and IDAs (*Fabczak and Osinka, 2019*), but a recent study suggests that distinct DNAAFs may be dedicated for the assembly of specific sub-sets of dyneins (*Yamaguchi et al., 2018*). Another recent study has suggested that distinct DNAAFs may catalyze specific stages of the dynein assembly process (*Mali et al., 2018*). This complexity motivates another key question, this one regarding the spatial organization of axonemal dynein assembly.

Several studies indicated that the DNAAFs and chaperones act together in cytosolic foci (*Diggle et al., 2014*; *Horani et al., 2012*; *Li et al., 2017*; *Mali et al., 2018*), and we recently demonstrated that DNAAFs, chaperones, and axonemal dynein subunits are sequestered in specialized membrane-less organelles that we termed DynAPs (*Huizar et al., 2018*). DynAPs are multiciliated cell (MCC)-specific organelles that display hallmarks of biological phase separation, with DNAAFs and chaperones fluxing through rapidly while dynein subunits are stably retained (*Huizar et al., 2018*). Though DynAPs display properties similar to many ubiquitous liquid-like organelles (*Banani et al., 2017*; *Shin and Brangwynne, 2017*), little else is known of the cell biology underlying DynAP assembly or function.

We show here that DynAPs contain functionally distinct sub-compartments. Confocal imaging revealed that the organelles defined by enrichment of DNAAFs and/or chaperones are sub-divided into sub-compartments specifically enriched for ODA or IDA subunits. Affinity-purification and mass-spectrometry of IDA and ODA subunits from MCCs identified several novel interactors, including the uncharacterized protein Daap1/C16orf71. We show that Daap1 is a cytoplasmic protein that is enriched in DynAPs in both human and *Xenopus* MCCs. Moreover, Daap1 localization is restricted to the ODA sub-region of DynAPs, where assays of fluorescence recovery after photobleaching (FRAP) show it is stably retained. Finally, disruption of Daap1 elicits a severe loss of ODAs from motile cilia. These data provide new insights into the structure and function of a still poorly defined, cell-type specific, disease-associated organelle.

## Results

### Outer and inner arm dynein subunits occupy sub-regions within DynAPs

The *Xenopus* embryo epidermis is a mucociliary epithelium that serves as a highly tractable model for understanding motile cilia (*Hayes et al., 2007*; *Walentek and Quigley, 2017*; *Werner and Mitchell, 2012*). Using this system, we previously showed that while DNAAFs, chaperones, and dyneins all co-localize in DynAPs with the canonical DNAAF, Ktu/Dnaaf2, dynein subunits displayed relatively more variable co-localization (*Huizar et al., 2018*). Here, we explored these localization patterns in more detail by pairwise co-expression of several fluorescently tagged DNAAFs, Hsp chaperones, and axonemal dynein subunits. We examined their localization in *en face* optical sections through the cytoplasm of *Xenopus* MCCs in vivo using confocal microscopy (*Figure 1A*, lower inset).

We found, for example, that the Hsp co-chaperone Stip1/Hop displayed essentially total co-localization with the assembly factor Dnaaf5/Heatr2 (*Figure 1B*), while by contrast, the outer dynein arm subunit Dnai2 clearly displayed only partial overlap with Dnaaf5 (*Figure 1C*). In fact, pairwise tests of five distinct DNAAFs and three chaperones revealed consistent, strong co-localization in all combinations (*Figure 1D,E,G*). Conversely, pairwise tests with three different axonemal dynein subunits consistently revealed only partial co-localization (*Figure 1F,G*). In all cases, IDA or ODA subunits occupied distinct, well-demarcated sub-regions within DNAAF- or chaperone-labeled DynAPs

(*Figure 1F*). Pearson correlations revealed that these differences were statistically significant (*Figure 1—figure supplement 1*).

We then performed pairwise tests of co-localization of several axonemal dynein subunits. Strikingly, we found that IDAs and ODAs were partitioned into mutually exclusive sub-regions. For example, the ODA subunit Dnai2 displayed near-perfect co-localization with two additional ODA subunits, Dnai1 and Dnal4 (*Figure 2A,B*). By contrast, Dnai2 displayed very little colocalization with IDA-*f* subunit Wdr78 or the IDA-*a, c, d* subunit Dnali1 (*Figure 2C,D*; *Figure 2—figure supplement 1A, B*). Again, Pearson correlations revealed these differences to be highly significant (*Figure 2—figure supplement 1C*).

### Specific sub-classes of inner dynein arms are partitioned to distinct sub-compartments within DynAPs

We further observed that the different sub-classes of IDAs were also restricted to discrete sub-regions within DynAPs. For example, the IDA-*a, c, d* subunit Dnali1 and the IDA-*f* subunit Wdr78 displayed only very little co-localization (*Figure 2E*; *Figure 2—figure supplement 1D*). By contrast, Wdr78 displayed significantly stronger co-localized with another IDA-*f* subunit, Tctex1d2 (*Figure 2F*; *Figure 2—figure supplement 1D*).

Together, these data suggest that axonemal dynein subunits are partitioned into functionally distinct sub-regions; based upon their contents, with ODAs, and at least two sub-classes of IDAs

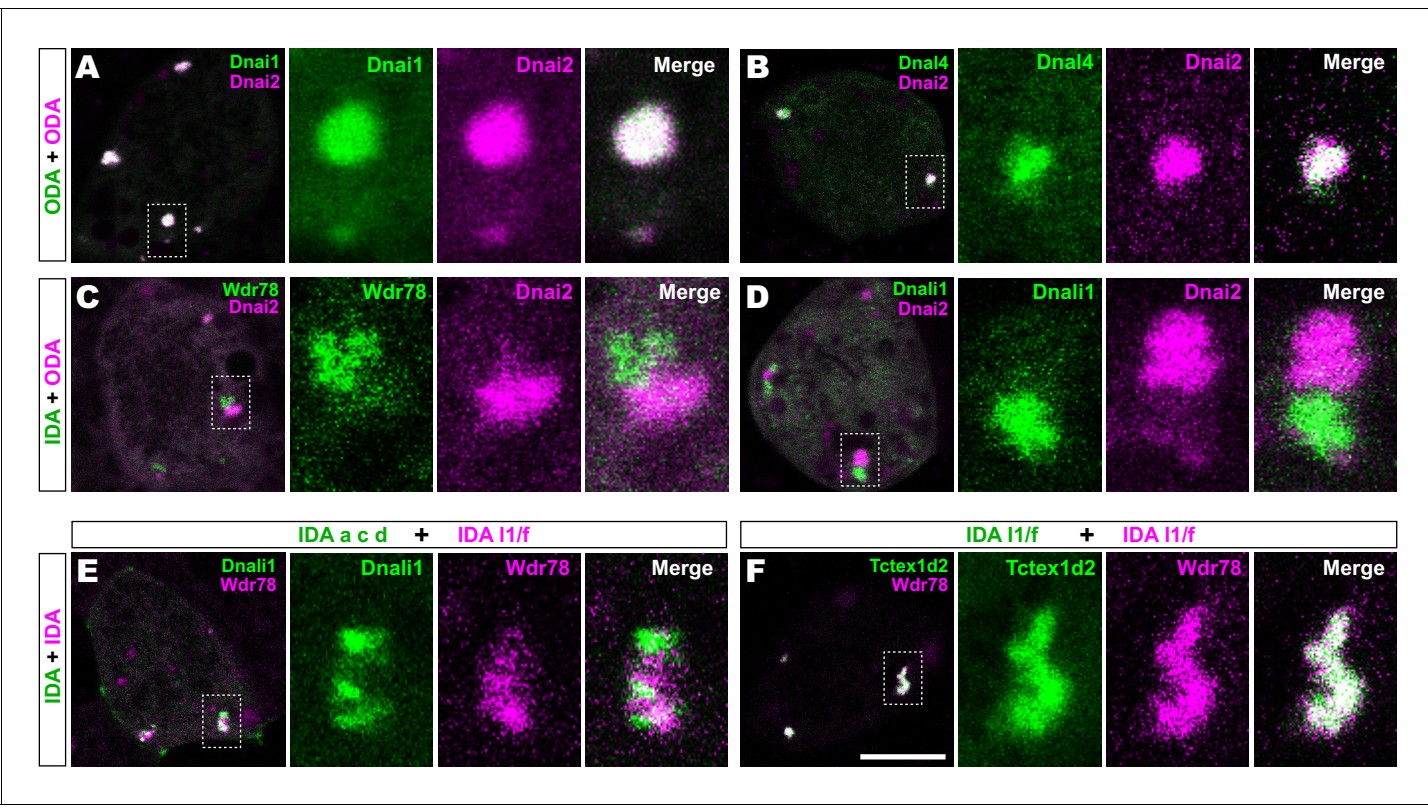

**Figure 2.** Outer and inner dynein arms localize to distinct sub-regions in DynAPs. (**A**) *En face* optical section of a *Xenopus* MCC expressing ODA subunits GFP-Dnai1 and mCherry-Dnai2 (as per lower inset in *Figure 1A*). The dashed box indicates the area shown at higher magnification in the panels to the right. Note near complete co-localization. (**B**) mCherry-Dnai2 also displays near-total co-localization with another ODA subunit, Dnal4. (**C, D**) mCherry-Dnai2 shows very little co-localization with the IDA subunit GFP-Wdr78 or GFP-Dnali1. (**E**) IDA-f subunit mCherry-Wdr78 displays little co-localization with IDA-a, c, d subunit GFP-Dnali1. (**F**) IDA-f subunits, mCherry-Wdr78 and GFP-Tctex1d2, show complete co-localization. Scale bar = 10 µm.

The online version of this article includes the following figure supplement(s) for figure 2:

**Figure supplement 1.** Co-localization data for IDAs and ODAs in DynAPs.

occupying functionally-related restricted spaces within DynAPs. We will refer to these here as 'ODA sub-DynAPs' and 'IDA-f sub-DynAPs,' etc.

## Affinity-purification and mass-spectrometry identifies specific in vivo interactors for vertebrate inner and outer dynein arm subunits in *Xenopus* MCCs

Given the essential role of cytoplasmic DNAAFs and chaperones in assembly of axonemal dyneins (*Desai et al., 2018*; *Fabczak and Osinka, 2019*), we reasoned that sub-DynAPs may also contain regulatory factors with specific roles in the assembly or deployment of either IDAs or ODAs. To test this idea, we sought to determine the interactomes of IDAs and ODAs directly in MCCs. We therefore developed a method to perform affinity-purification and mass-spectrometry (APMS) of in vivo interaction partners using *Xenopus* mucociliary epithelium and then examined the localization of interaction partners using in vivo imaging.

For this experiment, we took advantage of organotypically cultured explants of pluripotent ectoderm from early *Xenopus* embryos (so-called 'animal caps'), which can be differentiated into a wide array of organs and tissues, including mucociliary epithelium (*Ariizumi et al., 2009*; *Walentek and Quigley, 2017*; *Werner and Mitchell, 2012*). We and others have used such explants previously for large-scale genomic studies of ciliogenesis and cilia function (*Chung et al., 2014*; *Kim et al., 2018*; *Ma et al., 2014*; *Quigley and Kintner, 2017*; *Walentek et al., 2016*).

As baits for APMS, we used GFP-fusions to either IDA or ODA subunits, expressing them specifically in MCCs using a well-defined MCC-specific α-tubulin promoter (*Deblandre et al., 1999*; *Tu et al., 2018*). For each experiment, we excised ~550 animal caps and cultured them until MCCs developed beating cilia (NF stage 23)(*Figure 3A*, green, blue arrows). Protein was isolated from these tissue explants and APMS was performed using an anti-GFP antibody (*Figure 3A*). To control for non-specific interactions, each APMS experiment was accompanied by a parallel APMS experiment (i.e. from explants cut from the same clutch of embryos) using un-fused GFP (*Figure 3A*, gray). The results of the latter experiment were then used to subtract background (*Figure 3A*, right; see Materials and methods), and we calculated a Z-test, fold-change, and a false discovery rate for differential enrichment of identified proteins (see Materials and methods).

To identify ODA interactors, we used GFP-Dnai2; for IDAs we used GFP-Wdr78 (*King, 2018*).

For both proteins, we found that the bait itself was the most strongly enriched hit in the APMS, providing an important positive control (Circled in *Figure 3—figure supplement 1C, E*; see *Supplementary files 1*, *2*). Dnai2 pulldown strongly enriched for known ODA-specific sub-units, including the intermediate chain Dnai1, the heavy chains Dnah9/11, and the light chain Tctex1d1, among others (*Figure 3B*; *Figure 3—figure supplement 1A–C*; *Supplementary file 1*). Conversely, our Wdr78 sample was enriched for IDA components, specifically the IDA-*f* subtype, as expected (*Figure 3C*); these included the axonemal dynein heavy chains Dnah2 and Dnah10, as well as several light and intermediate chains (*Figure 3C*; *Figure 3—figure supplement 1D–F*; *Supplementary file 2*).

Finally, we identified weaker but significant interaction of Dnai2 and Wdr78 with a variety of known chaperones and assembly factors, including Ruvbl1/2, Spag1, and Dnaaf1 (*Supplementary files 1*, *2*), consistent with findings that these proteins are enriched in DynAPs and are essential for axonemal dynein assembly (*Fabczak and Osinka, 2019*; *Huizar et al., 2018*). Our data provide new insights into the compositions of ODAs and *f*-type IDAs in *Xenopus* MCCs and also serve as positive controls for the specificity of our APMS approach in MCCs.

## Identification and localization of novel IDA and ODA interactors

Importantly, our APMS identified not only known interactors, but also several novel interaction partners. For example, we observed interaction between Dnai2 and Nme9 (*Figure 3B*). Nme9 is highly similar to the *Chlamydomonas* ODA light chain LC3 and is present in the axonemes of airway cilia and sperm flagella in mice (*Sadek et al., 2003*). As in mice, *Xenopus* GFP-Nme9 localized to MCC axonemes (*Figure 4A*). More importantly, however, we also observed GFP-Nme9 in the cytoplasm, where it colocalized with Dnai2 specifically in ODA sub-DynAPs (*Figure 4B,C*).

Another notable interaction was that between Wdr78 and Wdr18 (*Figure 3C*). This result is significant because Wdr18 has been implicated in the control of ciliary beating, but its mechanism of

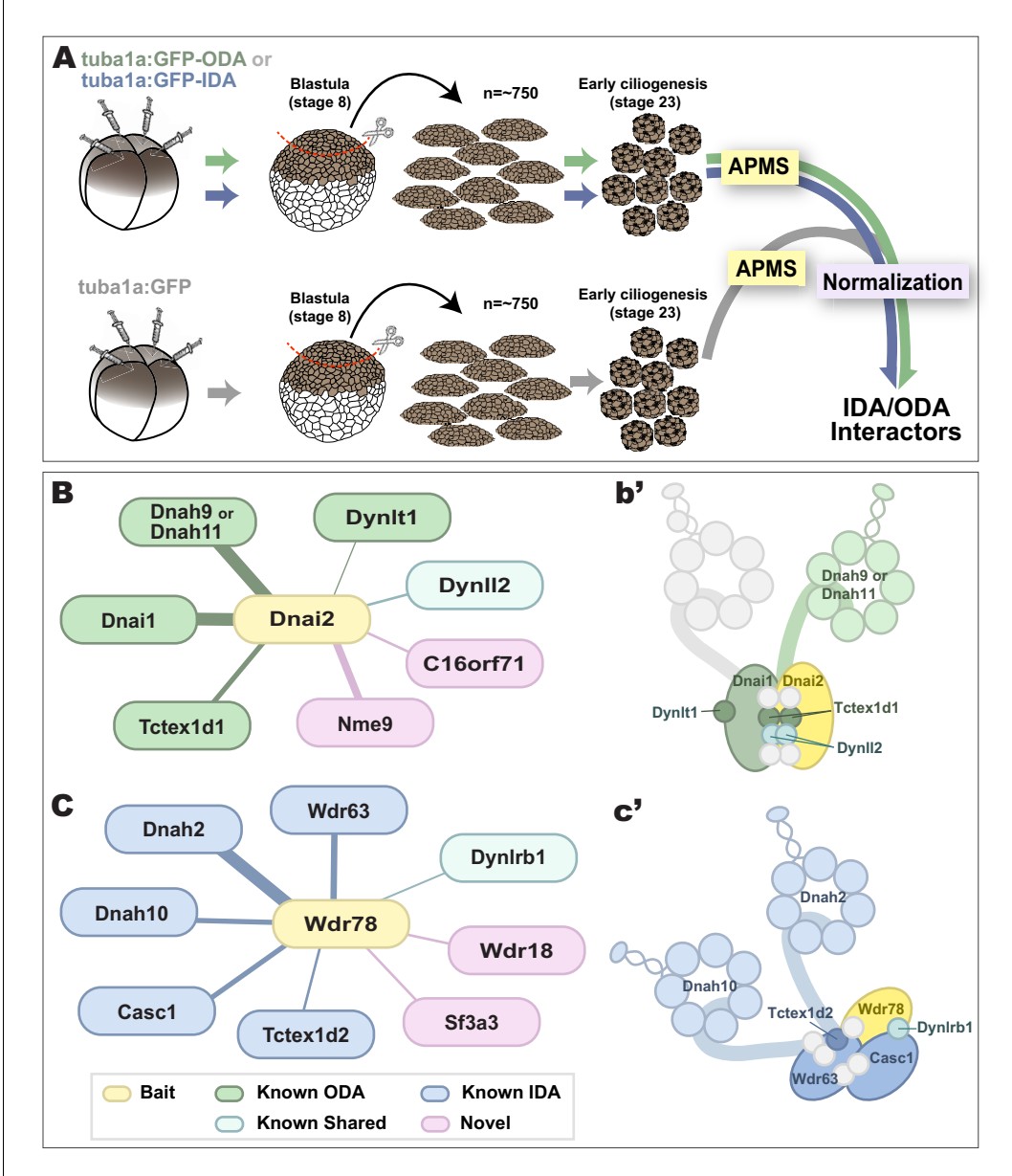

**Figure 3.** Specific identification of outer and inner arm dynein interactors. (**A**) Schematic of APMS workflow for identifying in vivo ODA and IDA interactors. GFP-tagged ODA or IDA driven by MCC-specific alpha-tubulin promoter by plasmid injection into *Xenopus* embryos and animal caps were isolated at stage 8. The cultured explants were collected at early ciliogenesis stage (stage 23) and subjected to APMS. Unfused GFP was assessed simultaneously and the data subtracted to control for non-specific interactions. (**B**) Spoke diagram displaying Dnai2 (ODA) interactors, line weight of spokes indicates Log2 fold-change of PSMs. b'. Schematic of outer dynein arm indicating identified Dnai2 preys. (**C**) Spoke diagram displaying Wdr78 (IDA) interactors. c'. Schematic of inner dynein arm (f type) indicating identified Wdr78 preys.

The online version of this article includes the following figure supplement(s) for figure 3:

**Figure supplement 1.** Proteins co-precipitating with Dnai2, Wdr78, and C16orf71 by APMS.

action is entirely unknown (*Gao et al., 2011*; *Silversides et al., 2012*). Interestingly, GFP-Wdr18 was not present in the axonemes of motile cilia (not shown) but did localize to DynAPs (*Figure 4D*). Interestingly, GFP-Wdr18 was also strongly enriched in the apical cytoplasm of MCCs, near the basal bodies (*Figure 4E*). Similarly, another IDA interactor, Sf3a3 (*Figure 3C*), was also absent from

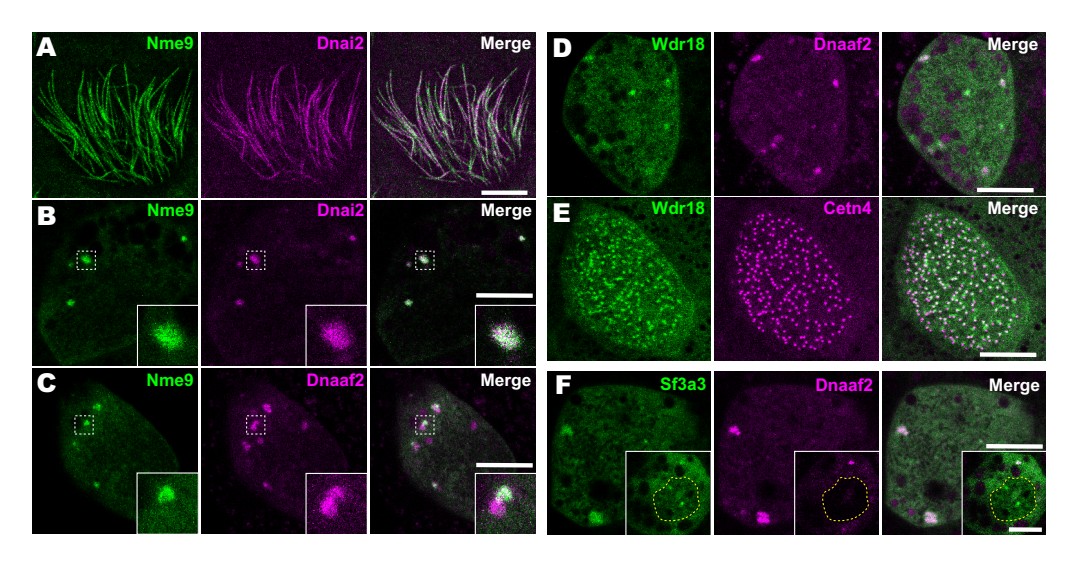

**Figure 4.** Localization of novel IDA and ODA interactors. (**A**) *En face* optical section just above the apical surface of a *Xenopus* MCC showing axonemal localization of GFP-Nme9. (**B**) A similar optical section through the MCC cytoplasm shows near-perfect colocalization of GFP-Nme9 with the ODA subunit mCherry-Dnai2 in DynAPs. Dashed box indicates area shown at higher magnification in the inset. (**C**) GFP-Nme9 only partially co-localizes with mCherry-Dnaaf2. (**D**) *En face* optical section through the cytoplasm shows colocalization of GFP-Wdr18 with mCherry-Dnaaf2 in DynAPs. (**E**) A similar section just below the MCC apical surface reveals GFP-Wdr18 localization near basal bodies labeled with RFP-Centrin4 (**F**) GFP-Sf3a3 co-localizes with mCherry-Dnaaf2. Scale bars = 10 μm.

The online version of this article includes the following figure supplement(s) for figure 4:

**Figure supplement 1.** Sf3a3-GFP (green) also localizes to speckles within the nucleus (labeled with RFP-Histone2B, red) in a *Xenopus* MCC.

axonemes (not shown), but did localize in DynAPs (*Figure 4F*). This protein was previously implicated in splicing (*Krämer et al., 2005*), so as expected, we also observed GFP-Sf3a3 in nuclear foci in *Xenopus* MCCs (*Figure 4—figure supplement 1*). Together, these data demonstrate that our APMS approach in *Xenopus* effectively identified novel, biologically plausible IDA and ODA-interacting proteins in vivo.

## C16orf71/Daap1 is a novel ODA-interacting protein with an interesting evolutionary trajectory

The most interesting interaction we identified in this study was that between Dnai2 and C16orf71 (*Figure 3B*). Though highly conserved among vertebrates, this protein has yet to be the subject of even a single published study. We first sought to confirm that C16orf71 is a *bona fide* ODA interactor using a reciprocal APMS experiment, with GFP-tagged *Xenopus* C16orf71 as the bait. This analysis revealed interaction of C16orf71 not only with Dnai2, but also other ODA subunits such as Dnai1, Dnah9 and Nme9, as well as Ruvbl1 (*Figure 5A*; *Figure 3—figure supplement 1G–I*, Supp. File 3). Based on these findings and data described below, we propose to rename this protein Daap1, for Dynein Axonemal-Associated Protein 1.

In situ hybridization revealed that *Xenopus Daap1* expression was enriched in MCCs in the epidermis and the nephrostomes of the pronephros (*Figure 5B*). Likewise, examination of data from the Human Protein Atlas (*Uhlén et al., 2015*) revealed that human DAAP1 protein is present exclusively in tissues with motile cilia (*Figure 5C*). Interestingly, while Daap1 is conserved among vertebrates, it appears to have followed an interesting evolutionary trajectory. In mammals, including humans, orthologs of Daap1 are composed entirely of a domain of unknown function (DUF4701), which phylogenetic analysis suggests is vertebrate-specific (*Figure 5D*). Our analysis of this sequence suggests it is intrinsically disordered (*Figure 5E*), a feature commonly associated with liquid-like organelles. By contrast, all non-mammalian vertebrate Daap1 orthologs contain not only the DUF4701 domain, but also a long C-terminal extension harboring, remarkably, an NDK domain

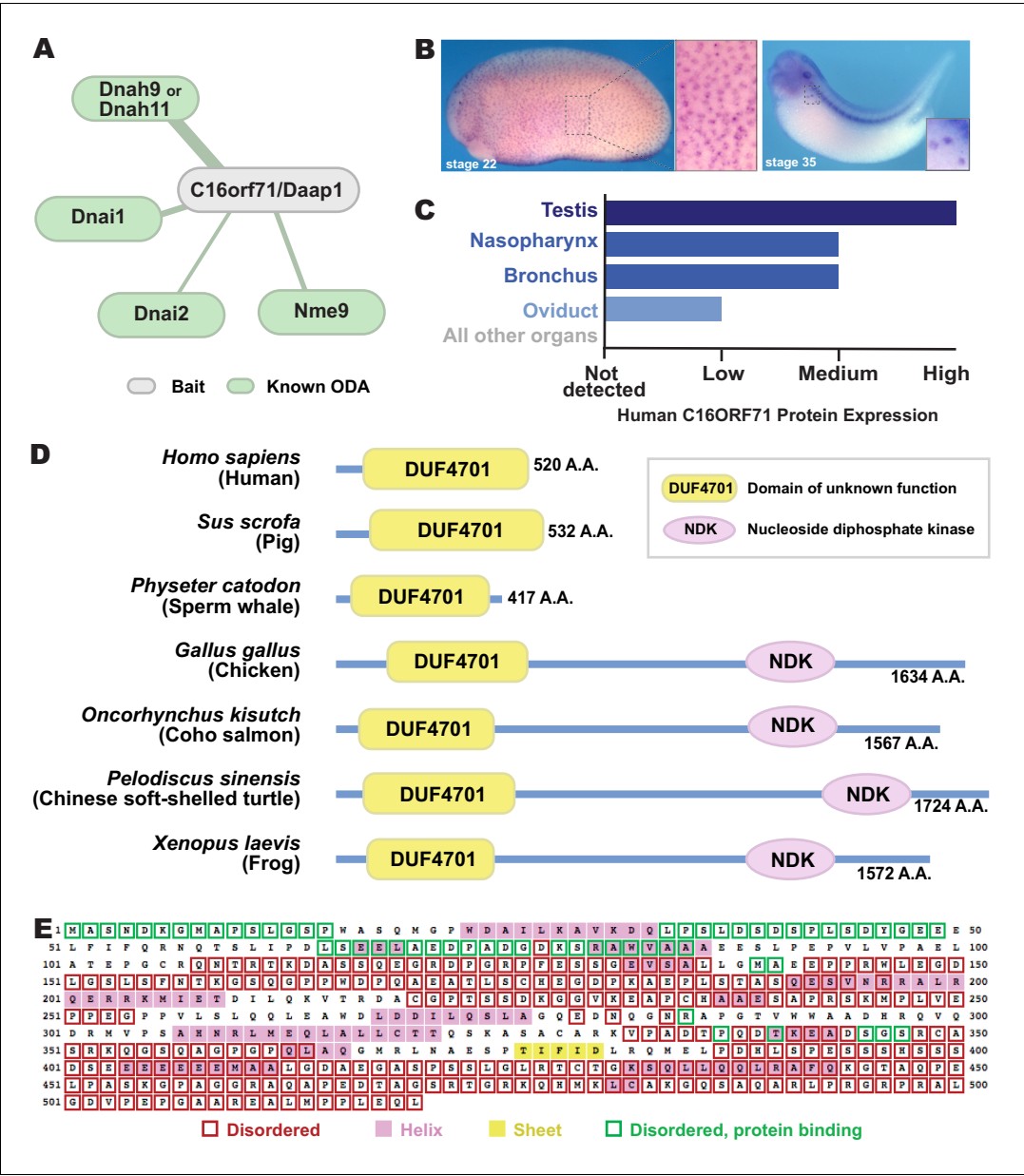

**Figure 5.** C16orf71/Daap1 is a novel ODA interactor specifically expressed in MCCs. (**A**) Spoke diagram displaying C16orf71/Daap1 interactors; line weight indicates Log2 fold- change of PSMs. (**B**) In situ hybridization of *Daap1* in *Xenopus* reveals expression in epidermal MCCs (left) and MCCs in the nephrostomes (right). (**C**) Graph showing C16orf71 protein expression levels in human tissues from the Human Protein Atlas (*Uhlén et al., 2015*). (**D**) Domain organization of C16orf71/Daap1 orthologs across vertebrates from NCBI Orthologs. (**E**) Domain prediction with human C16orf71/DAAP1 sequence from the PSIPRED Protein Analysis Workbench for disorder and secondary structure prediction.

similar to those in Nme8 and Nme9 (*Figure 5D*). We therefore chose this novel ODA interactor for more in-depth study.

## Daap1 is a DynAP-specific protein in *Xenopus* and human MCCs

We next examined the localization of the Daap1 protein in *Xenopus* MCCs. Intriguingly, no signal was detectable in MCC axonemes as marked by Dnai2, and instead, GFP-Daap1 was localized exclusively to foci in the cytoplasm of MCCs (*Figure 6A,B*). GFP-Daap1 strongly co-localized with Dnai2 (*Figure 6B*), suggesting that the interaction we identified by APMS occurs in DynAPs. Indeed,

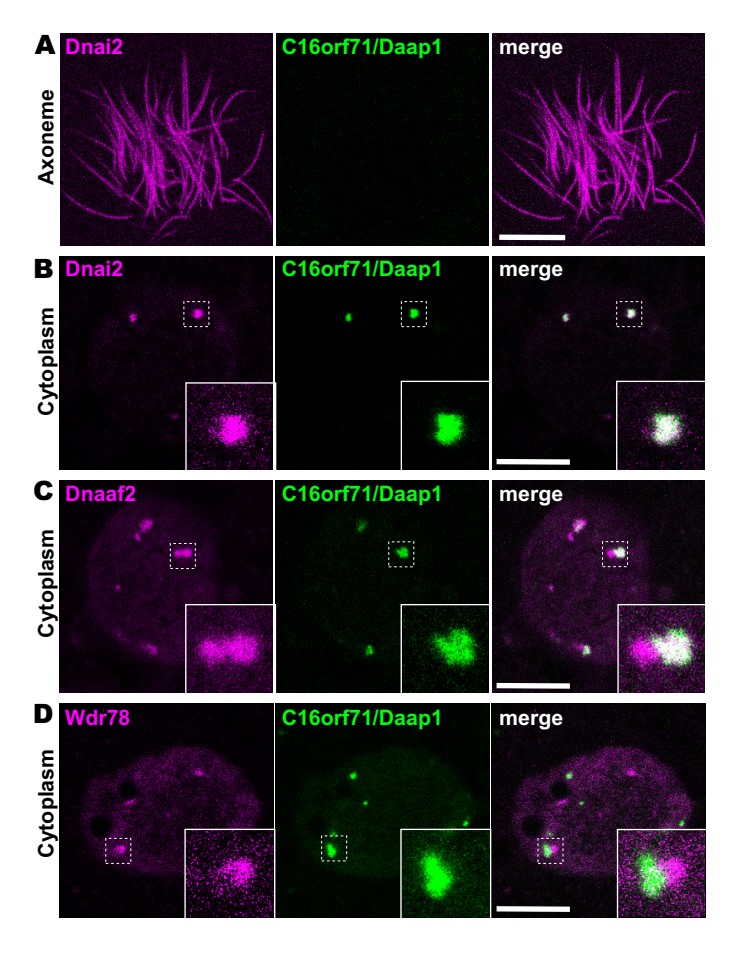

**Figure 6.** Daap1 is a DynAP-specific protein in Xenopus MCCs. (**A**) *En face* optical section above the apical surface of a *Xenopus* MCC reveals absence of GFP-Daap1 in axonemes. (**B**) A similar optical section through the MCC cytoplasm shows near-perfect colocalization of GFP-Daap1 with the ODA subunit mCherry-Dnai2 in DynAPs. Inset shows higher magnification view of dashed box for each panel. (**C**) GFP-Daap1 only partially co-localizes with mCherry-Dnaaf2. (**D**) GFP-Daap1 co-localizes only very weakly with the IDA subunit mCherry-Wdr78. Scale bars = 10 µm.

Daap1 displayed a partial co-localization with Dnaaf2 similar to that observed in the cytoplasm for other ODA subunits (*Figure 6C*). Finally, Daap1 displayed little or no overlap with GFP-Wdr78 (*Figure 6D*), suggesting that it localized specifically in ODA sub-DynAPs (*Figure 6D*).

Because human DAAP1 lacks the C-terminal NDK domain present in the *Xenopus* ortholog (*Figure 5D*), we were curious to know if the human ortholog also localizes to DynAPs. We therefore performed immunostaining for DAAP1 on sections of human lung. As observed in *Xenopus*, DAAP1 was not detectable in the axonemes of human MCCs (*Figure 7A*, brackets), and cytoplasmic staining for DAAP1 was strongly enriched in DynAPs, as indicated by co-labeling for the ODA subunit DNAI1 (*Figure 7A*, arrowheads).

This result then prompted us to examine the localization of a deletion construct comprising the isolated N-terminal DUF4701 domain of *Xenopus* Daap1 (i.e. a *Xenopus* equivalent of human DAAP1)(*Figure 7B*). We found that GFP-Daap1-Nterm also localized to ODA sub-DynAPs in *Xenopus* MCCs (*Figure 7C–E*). Thus, despite their divergent evolutionary trajectories, both *Xenopus* and human DAAP1 are DynAP-specific cytoplasmic proteins, and the N-terminal DUF4701 domain is sufficient to direct localization to ODA sub-DynAPs.

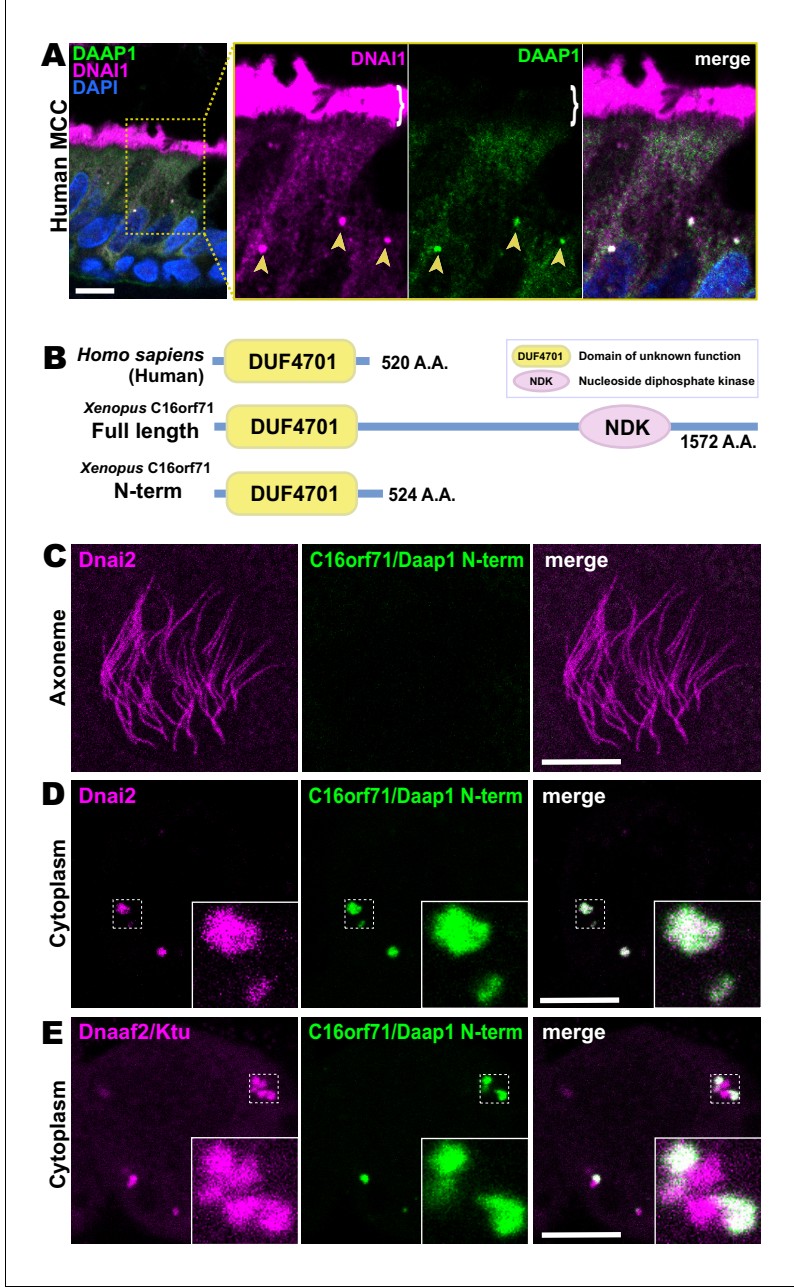

**Figure 7.** Human DAAP1 is a DynAP-specific protein. (**A**) A primary human lung section immunostained for human DAAP1 (green) and the ODA subunit DNAI1 (magenta). Boxed area indicates magnified region shown at right. DNAI1 strongly labels MCC axonemes (bracket) and also DynAPs in the cytoplasm (arrowheads). As in *Xenopus*, human DAAP1 is not present in axonemes but is strongly enriched in DynAPs. (DAPI (blue) marks nuclei; scale bar = 10 μm) (**B**) Schematic of C16orf71 constructs: *Xenopus* Daap1-Nterm is truncated, containing only the DUF4701 domain, similar to human DAAP1. (**C**) *En face* optical section above the apical surface of a *Xenopus* MCC reveals absence of Daap1-Nterm-GFP in axonemes. (**D, E**) Similar optical sections through the cytoplasm shows near-perfect colocalization of GFP-Daap1-Nterm with the ODA subunit mCherry-Dnai2 and only partial co-localization with Dnaaf2/Ktu. In all cases, inset shows higher magnification view of dashed box for accompanying panels. Scale bars = 10 μm.

## Daap1 is stably retained in ODA sub-DynAPs

DynAPs are liquid-like organelles, and we previously found using FRAP that both the DNAAFs and broadly-acting chaperones flux rapidly through these organelles, while both ODA and IDA subunits are stably retained (*Huizar et al., 2018*). Because Daap1 does not localize to axonemes in either humans or *Xenopus*, its localization to DynAPs suggests a role in ODA processing or assembly, rather than a direct function in ciliary beating. We therefore expected Daap1 to flux rapidly through DynAPs, similar to other cytoplasmic assembly factors. To our surprise, however, FRAP revealed instead that *Xenopus* Daap1 was retained in DynAPs in a manner that was similar, but not identical, to dynein subunit Dnai2 (*Figure 8*). Specifically, Daap1 displayed a somewhat slower initial recovery, but rather than plateauing at around 20 s, the signal continued to slowly recover for at least 60 s (*Figure 8B*).

We were also interested to know if human Daap1 displays similar dynamics, but FRAP assays in human MCCs are very challenging. We therefore examined the FRAP kinetics of *Xenopus* Daap1-Nterm as a proxy of the human protein for use in *Xenopus* MCCs. In FRAP assays, Daap1-Nterm

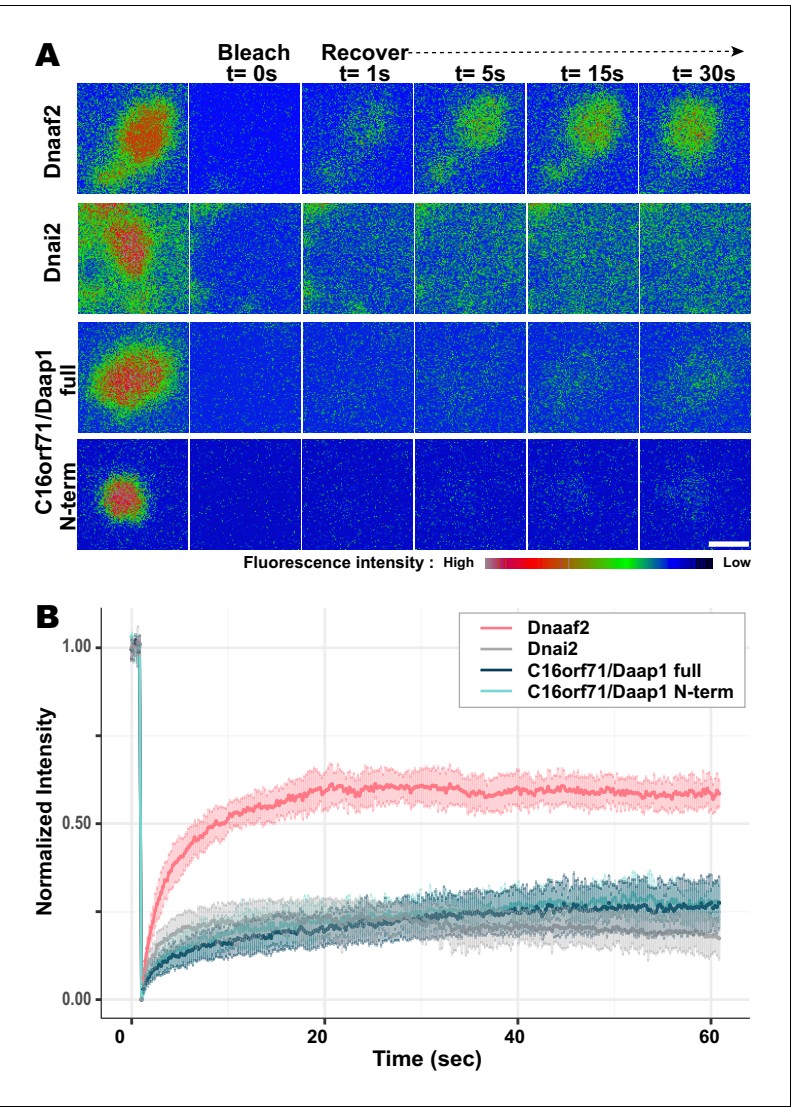

**Figure 8.** Daap1 is stably retained in DynAPs.  (**A**) Stills from time-lapse imaging of FRAP for the assembly factor GFP-Dnaaf2, the ODA subunit GFP-Dnai2, GFP-Daap1 and GFP-Daap1-Nterm. Images are color-coded to highlight changes in pixel intensity (see key below images). Scale bar = 1 μm (**B**) Graphs displaying FRAP curves for GFP-Dnaaf2 (n = 14), GFP-Dnai2 (n = 12) GFP-Daap1 (n = 26), and GFP-Daap1-Nterm (n = 22).

displayed essentially identical kinetics to full-length Daap1 (*Figure 8B*). This result is consistent with the localization data above and strongly suggests that the DUF4701 domain is sufficient to drive the retention of Daap1 in ODA sub-DynAPs.

## Daap1 is required for deployment of ODAs to the axoneme

We tested this idea using morpholino (MO)-mediated knockdown (KD) to disrupt splicing of the *Daap1* transcript in *Xenopus* (*Figure 9—figure supplement 1A*). Consistent with its interaction specifically with ciliary beating machinery, Daap1-KD had no impact on ciliogenesis itself, and well-defined ciliary tufts were observed in all morphants. However, Daap1-KD elicited severe loss of ODAs from MCC axonemes (*Figure 9A,B*, D, E). Both the intermediate chain Dnai2, as well as the light chain Dnal1 were severely reduced, with only small domains of the proximal axoneme labeled by either marker following knockdown (*Figure 9A,B*). We quantified this phenotype by examining intensity plots for Dnai2 and Dnai1 along the length of motile axonemes and normalizing this against co-expressed membrane-RFP (*Figure 9C,F*). Importantly, this phenotype was also observed using a second MO targeting a distinct splice site, suggesting that this phenotype is a specific effect of Daap1 depletion. (*Figure 9—figure supplement 1B,C*).

Given the specific localization of Daap1 to ODA sub-DynAPs, we expected that Daap1 loss should specifically impact axonemal deployment of ODAs, and for the most part, this was the case. While both ODA proteins were depleted along the entire length of the axoneme, the IDA-*a, c, d* protein Dnali1 and the IDA-*f* protein Wdr78 displayed more subtle defects (*Figure 9G,H,J,K*). Dnali1 displayed a mild reduction along the entire axoneme (*Figure 9I*), while Wdr78 displayed an even mild disruption that was largely restricted to the distal axoneme (*Figure 9L*). Interestingly, this result may relate to the partial co-localization of Dnali1 with Dnai2 (*Figure 2—figure supplement 1C*) and reflects the known intimate link between assembly of ODAs and IDA-*c* (e.g. *Yamaguchi et al., 2018*; *Yamamoto et al., 2020*).

Overall, comparisons of subunit intensities demonstrated that loss of Daap1 had a significantly greater impact on ODA compared to IDAs. Because the effect on IDA-*f* was quite subtle (*Figure 9L*), we also made a more direct comparison by generating intensity plots to directly compare Wdr78 and Dnai1 co-expressed in the same axonemes (*Figure 9M–P*). This analysis clearly demonstrated a profoundly more substantial loss of ODAs than IDAs after Daap1 depletion.

Thus, Daap1 is a DynAP-specific protein that interacts with ODA subunits and is essential for the deployment of ODAs to axonemes and further suggests a complex interplay of ODAs and IDAs is required to establish the final patterning of motors that drive ciliary beating.

## Discussion

Here, we have used live imaging to demonstrate that DynAPs contain sub-compartments that specifically partition inner and outer dynein arm subunits (*Figures 1* and *2*). Using in vivo proteomics, we identify several novel axonemal dynein interactors, including C16orf71/Daap1 (*Figures 3* and *5*). In both *Xenopus* and humans, Daap1 is present exclusively in the cytoplasm and not in ciliary axonemes. Moreover, the protein is highly enriched in DynAPs, where it is restricted to the ODA sub-DynAP (*Figures 6–8*). Remarkably, loss-of-function experiments revealed a requirement for Daap1 predominantly in the deployment of ODAs throughout the length of MCC cilia, while the impact on IDAs was only very mild, and restricted largely to the distal axoneme (*Figure 9*). Together, these data are significant for (1) adding weight to the argument that DynAPs serve a specific function in the assembly of axonemal dyneins, (2) demonstrating that sub-DynAPs represent molecularly and functionally distinct spaces within DynAPs, and (3) suggesting that additional DynAP-specific regulatory factors remain to be discovered.

### Novel axonemal dynein interactors

In addition to the mechanistic insights provided by this study (discussed in detail below), our method for APMS and the datasets of in vivo IDA and ODA interactors provided here will be a useful resource for future studies. For example, *wdr18* knockdown results in left/right asymmetry defects in zebrafish and mutations in human *WDR18* are associated with congenital heart defects (*Gao et al., 2011*; *Silversides et al., 2012*); both phenotypes are indicative of ciliary beating defects, but the mechanism of Wdr18 action is entirely unknown. The data here are significant, then, for connecting

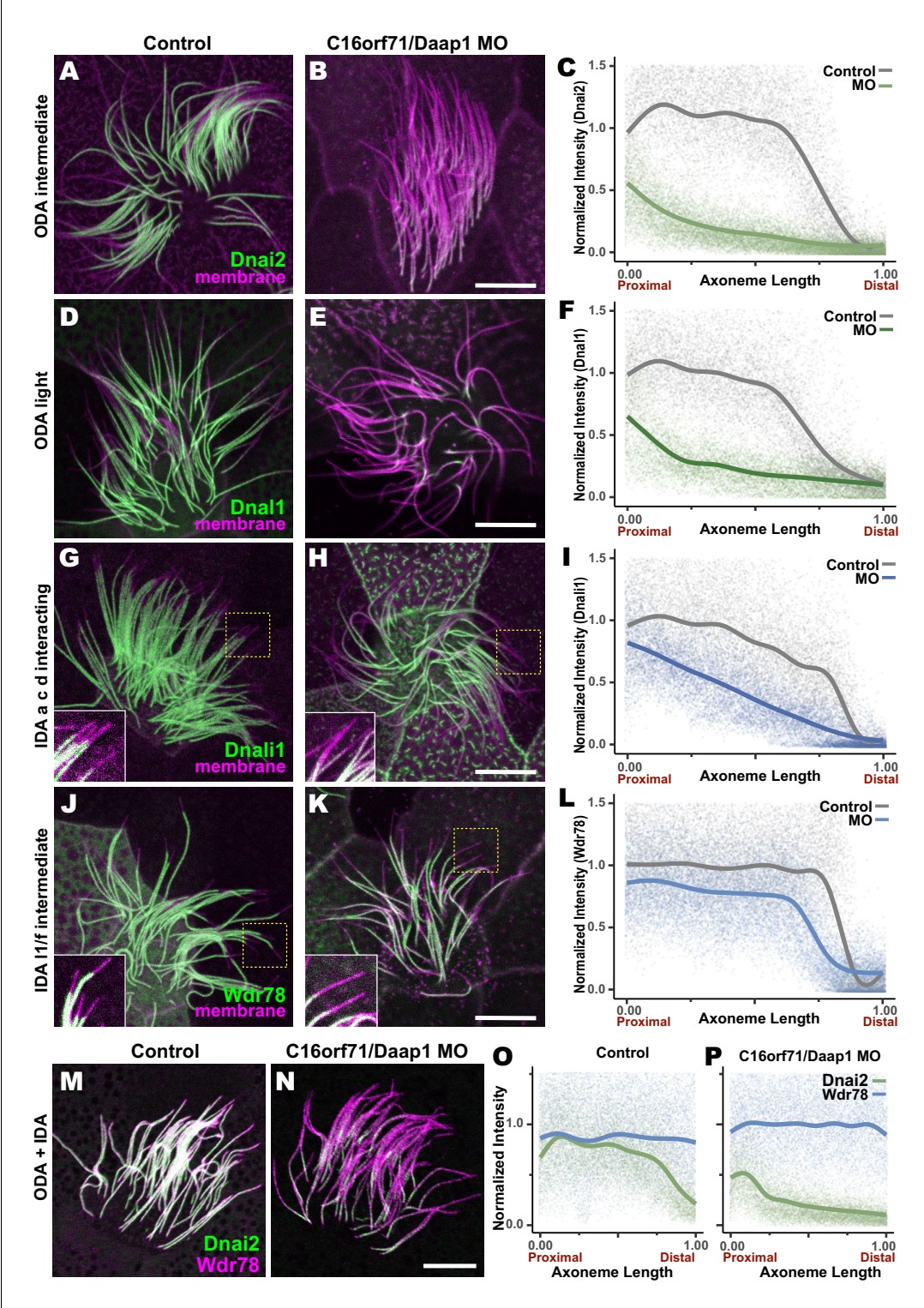

**Figure 9.** Loss of C16orf71 disrupts deployment of dynein subunits to the axoneme.  (A–B) *Xenopus* MCC axonemes labeled by membrane-RFP (magenta) together with the GFP-Dnai2, an ODA intermediated chain in control embryo (A) and in C16orf71/Daap1 morphant (B). (C) Graph showing intensity of GFP-Dnai2 along the normalized axoneme length (n = 29 for control, 51 for morphant). (D–E) MCCs labeled by membrane-RFP (magenta) together with the GFP-Dnal1, an ODA light chain in control embryo (D) and in C16orf71/Daap1 morphant (E). (F) Graph showing intensity of GFP-Dnal1 along the normalized axoneme length (n = 27 for control, 26 for morphant). (G–H) MCCs labeled by membrane RFP (magenta) together with the GFP-Dnali1, an IDA a, c, d interacting protein in control embryo (G) and in C16orf71/Daap1 morphant (H). (I) Graph showing intensity of GFP-Dnali1 along

*Figure 9 continued on next page*

*Figure 9 continued*

the normalized axoneme length (n = 30 for control, 33 for morphant). (J–K) MCCs labeled by membrane-RFP (magenta) together with the GFP-Wdr78, an IDA l1/f intermediated chain in control embryo (J) and in C16orf71/Daap1 morphant (K). (L) Graph showing intensity of GFP-Dnal1 along the normalized axoneme length (n = 32 for control, 43 for morphant). Both Dnai2, an ODA Intermediated chain and Dnal1, an ODA light chain are severely reduced in the axoneme in C16orf71 morphants, while IDA subunits display only mild loss from the distal most axoneme (inset). Yellow boxes indicate regions shown in accompanying insets for each panel. (M–N) MCCs co-expressing a marker for both ODAs (Dnai2, green) and IDAs (Wdr78, magenta). Loss of C16orf71 results in specific loss of ODAs in the axoneme (N), while both ODA and IDA properly localize in the axoneme in control (M). (O–P) Graphs showing intensities of GFP-Dnai2 and mCherry-Wdr78 along the normalized axoneme length in controls (O) and morphants (P). Scale bars = 10 μm.

The online version of this article includes the following figure supplement(s) for figure 9:

**Figure supplement 1.** Validation of C16orf71 morpholinos.

Wdr18 to IDAs (*Figure 3C*) and for reporting the localization of Wdr18 in motile ciliated cells for the first time. Because we found that Wdr18 was not present in the axonemes of motile cilia but was present in DynAPs and near basal bodies (*Figure 4D,E*), our data suggest that Wdr18 function may not relate directly to IDA *function*, but rather to the *assembly* or *transport* of IDAs.

The more curious result here is the interaction of Wdr78 with Sf3a3 (*Figure 3C*), a well-defined component of the RNA splicing machinery (*Krämer et al., 2005*). Though unexpected, this result, and our finding that Sf3a3 is enriched in DynAPs (*Figure 4F*) is consistent with our recent report that DynAPs contain both RNA and RNA-associated proteins (*Drew et al., 2020*), similar to other liquid-like organelles such as stress granules and P bodies (*Banani et al., 2017*; *Lin et al., 2015*; *Mittag and Parker, 2018*). Indeed, we found that Cfap43 and Cfap44, which tether IDA-*f* to micro-tubules (*Fu et al., 2018*; *Kubo et al., 2018*), are both RNA-associated and localized to sub-DynAPs in *Xenopus* MCCs (*Drew et al., 2020*).

These findings are especially interesting in light of two additional recent papers. First, a recent paper reports that splicing factor Srsf1 plays a *splicing-independent* role in ciliary beating, likely via control of translation of protein essential for ciliary beating (*Haward et al., 2020*). A second recent paper reports the presence of 'KL granules' in *Drosophila* sperm that -like DynAPs- contain the Dynein assembly chaperones Ruvbl1/2, and these granules are also highly enriched in the mRNAs encoding axonemal dyneins (*Fingerhut and Yamashita, 2020*). Taken together, these findings suggest that the role of RNA-associated proteins in DynAP assembly or function should be a rich area for future investigation.

## Daap1 and the Nme family of proteins and ciliary beating

Our data here also shed light on the evolution of motile cilia function and regulation in vertebrates, particularly with respect to Daap1 and the Nme family of NDK proteins (*Desvignes et al., 2009*). Three Nme proteins have been implicated in motile cilia function, including Nme8, which is ortholo-gous to the LC3 ODA subunit in *Chlamydomonas* and has been implicated in human motile ciliop-athy (*Duriez et al., 2007*; *Pazour et al., 2006*). Nme5 is also required for ciliary beating in *Xenopus* and mammals, though its mechanism of action remains unknown (*Anderegg et al., 2019*; *Chung et al., 2014*; *Vogel et al., 2012*). Nme9 (aka Txndc6 or Txl-2) is localized to motile cilia in the mouse (*Sadek et al., 2003*), but its function is also unknown. It is significant, then, that we found Nme9 to be localized to *Xenopus* MCC cilia (*Figure 4A–C*). The observed interactions with Dnai2 and Daap1 (*Figure 3C*, *Figure 5A*) suggest that it too may be an ODA subunit.

Curiously, examination of genome data using Xenbase (*Karimi et al., 2018*; *Session et al., 2016*) suggests that *Xenopus* lacks the Nme8 gene, and further exploration in NCBI revealed that many vertebrate genomes contain only Nme8 or Nme9, but not both (not shown). The linkage of an NDK domain to the DUF4701 domain in non-mammalian Daap1 orthologs suggest the possibility that these two protein domains may act in concert, and it is tempting to speculate that the NDK domain of Daap1 in non-mammalian vertebrates may explain the absence of either Nme8 or Nme9 in those genomes. Interestingly, a very recent report describes the discovery of novel NDK domain protein in *Tetrahymena* with a specific role in stabilizing ODA dyneins in an inactive conformation for targeting to axonemes (*Mali et al., 2020*). It is possible then that Daap1 plays a similar role in *Xenopus*. Con-tinued studies in diverse organisms will thus be of great interest.

## The composition and function of DynAPs

DynAPs have now been observed in human, mouse, *Xenopus* and zebrafish, so it is clear that these organelles represent a conserved element of MCCs (*Diggle et al., 2014*; *Horani et al., 2018*; *Huizar et al., 2018*; *Li et al., 2017*; *Mali et al., 2018*). However, the precise function of DynAPs remains unknown, a situation shared with many liquid-like organelles. Genetic evidence suggests that DynAPs are factories for axonemal dynein assembly, as genetic loss of DynAP-localized DNAAFs consistently results in a failure of dynein arm assembly, failure of dynein deployment to axonemes, and defective ciliary beating (*Fabczak and Osinka, 2019*). Our identification of Daap1 as yet another protein that is enriched in DynAPs and is specifically required for deployment of axonemal dyneins to cilia adds weight to the argument that DynAPs perform a critical assembly or deployment function.

Strictly speaking, however, all DNAAF proteins and chaperones yet studied are present through-out the cytoplasm and *enriched* in DynAPs, so we cannot rule out the possibility that dyneins are normally assembled in the cytoplasm, while DynAPs serve some other function. For example, DynAPs may act in storage or assembly of dyneins for rapid deployment to cilia (e.g. for ciliary regeneration) or for quality control and/or degradation of mis-folded dyneins. Answering such questions is a key challenge, and is one faced generally for liquid-like organelles.

Regardless, our data clearly demonstrate that DynAPs are structurally partitioned, as are many other liquid-like organelles (e.g. *Feric et al., 2016*; *Jain et al., 2016*; *Schmidt and Rohatgi, 2016*). Moreover, our observation that Daap1 is restricted to ODA sub-DynAPs (*Figures 6* and *7*) and that its loss results predominantly in ODA defects (*Figure 9*) argues that sub-DynAPs reflect a *functional* sub-compartmentalization. Conversely, though it has been suggested that DNAAFs and chaperones act in a relay system to catalyze iterative steps of ODA and IDA assembly (*Mali et al., 2018*), we found no evidence of physical compartmentalization of any of the tested DNAAFs and chaperones (*Figure 1*), suggesting that partitioned localization of those factors is not strictly required. Consistent with this idea, though Ktu has been shown to specifically impact assembly of ODAs and IDA-*c* (*Yamaguchi et al., 2018*), we found no restriction of Ktu localization within DynAPs (*Figure 1* and see *Huizar et al., 2018*). Together, our findings here provide a deeper cell biological framework in which to understand the complex genetics of dynein assembly in motile ciliated cells.

# Materials and methods

## *Xenopus* embryo manipulations

*Xenopus* embryo manipulations were carried out using standard protocols. Briefly, female adult *Xenopus* were induced to ovulate by injection of hCG (human chorionic gonadotropin). In vitro fertilization was carried out by homogenizing a small fraction of a testis in 1X Marc's Modified Ringer's (MMR). Embryos were dejellied in 1/3x MMR with 2.5%(w/v) cysteine (pH7.8). Embryos were microinjected with mRNA, circular DNA or morpholinos in 2% Ficoll (w/v) in 1/3x MMR and injected embryos were washed with 1/3x MMR after 2 hr.

## Plasmids and MOs for microinjections

*Xenopus* gene sequences were obtained from Xenbase (http://www.xenbase.org) and open reading frames (ORF) of genes were amplified from the *Xenopus* cDNA library by polymerase chain reaction (PCR), and then are inserted into a pCS10R vector containing a fluorescence tag. In addition to the vectors described previously (*Huizar et al., 2018*), the following constructs were cloned into pCS vector: ruvbl2-mCherry, mCherry-zmynd10, GFP-wdr78, GFP-dnai1, GFP-dnal4, GFP-dnal1, GFP-nme9, GFP-wdr18, GFP-sf3a3, GFP-c16orf71, GFP-c16orf71 N-term (1-527aa), GPF-tctex1d2 and mCherry-wdr78. These constructs were linearized and the capped mRNAs were synthesized using mMESSAGE mMACHINE SP6 transcription kit (ThermoFisher Scientific). Each 80 pg of each mRNA was injected into two ventral blastomeres. For APMS, GFP-dnai2, GFP-wdr78 and GFP-c16orf71 N-term were inserted into pCS2 vectors under multiciliated-cell-specific alpha-tubulin promoter and each 40 pg of each DNA was injected into blastomeres. C16orf71/Daap1 morpholinos were designed to target 1st or 3rd exon-intron splicing junction, The MO sequences and the working concentrations include:

MO #1: 5'-AGTAAGGTCTGTACACTTACCAGGG-3', 30 ng per injection
MO #2: 5'-ACAAATGCAAGTTTTTCTTACCTCA-3', 10 ng per injection

## Immunoprecipitation of *Xenopus* animal caps for mass-spectrometry

To identify Dnai2 or Wdr78 interactors, circular plasmids of GFP only, GFP-dnai2 and GFP-wdr78 driven by MCC-specific α-tubulin promoter were injected into 4-blastomeres of 4 cell stage *Xenopus* embryos. Approximately 550 animal caps per sample were isolated at stage eight using forceps and were cultured in 1X Steinberg's solution (0.58 mM NaCl, 0.64 mM KCl, 0.33 mM Ca(NO$_2$)$_2$, 0.8 mM MgSO$_4$, 5 mM Tris, 50 μg/ml gentamicin, pH 7.4–7.6) until sibling embryos reached stage 23. The cultured explants were collected and immunoprecipitation (IP) was performed using GFP-Trap Aga-rose Kit (ChromoTek, cat# gtak-20). Immunoprecipitated proteins were eluted in 2X sample buffer. For MS of C16orf71, GFP only and GFP-c16orf71 plasmid driven by alpha-tubulin promoter were injected into *Xenopus* embryos and ~500 animal caps were collected from each sample. The follow-ing steps were performed with the same procedures as described for Dnai2 and Wdr78 AP. For sta-tistical tests of enrichment, GFP-wdr78 and GFP-dnai2 share a GFP- only control (GFP_1a_08292018, GFP_1b_08292018) and GFP-c16orf71 uses a separate GFP-only control (GFP_1a_04252019, GFP_1b_ 04252019).

## Affinity-purification-mass-spectrometry

Immunoprecipitated proteins were resuspended in SDS-PAGE sample buffer and heated 5 min at 95°C before loading onto a 7.5% acrylamide mini-Protean TGX gel (BioRad). After 7 min of electro-phoresis at 100 V the gel was stained with Imperial Protein stain (Thermo) according to manufac-turer's instructions. The protein band was excised, diced to 1 mm cubes and processed by standard trypsin in-gel digest methods for mass-spectrometry (for experiments with baits Dnai2 and Wdr78) or by the more rapid method of *Goodman et al., 2018* for C16orf71. Digested peptides were desalted with HyperSep Spin Tip C-18 columns (Thermo Scientific), dried, and resuspended in 30–60 μl of 5% acetonitrile, 0.1% acetic acid for mass-spectrometry.

Samples from experiments with baits Dnai2 and Wdr78 were analyzed on a Thermo Orbitrap Fusion mass spectrometer and those with bait C16orf71 were analyzed using a Thermo Orbitrap Fusion Lumos Tribrid mass spectrometer. In all cases peptides were separated using reverse phase chromatography on a Dionex Ultimate 3000 RSLCnano UHPLC system (Thermo Scientific) with a C18 trap to Acclaim C18 PepMap RSLC column (Dionex; Thermo Scientific) configuration and eluted using a 3% to 45% gradient over 60 min. with direct injection into the mass spectrometer using nano-electrospray. Data were collected using a data-dependent top speed HCD acquisition method with full precursor ion scans (MS1) collected at 120,000 m/z resolution. Monoisotopic precursor selection and charge-state screening were enabled using Advanced Peak Determination (APD), with ions of charge $\geq$ + two selected for high energy-induced dissociation (HCD) with stepped collision energy of 30% +/- 3% (Lumos) or 31% +/- 4% (Orbitrap Fusion). Dynamic exclusion was active for ions selected once with an exclusion period of 20 s (Lumos) or 30 s (Orbitrap Fusion). All MS2 scans were centroid and collected in rapid mode.

Raw MS/MS spectra were processed using Proteome Discoverer (v2.3). We used the Percolator node in Proteome Discoverer to assign unique peptide spectral matches (PSMs) at FDR < 5% to the composite form of the *X. laevis* reference proteome described in *Drew et al., 2020*, which com-prises both genome-derived Xenbase JGI v9.1 + GenBank *X. laevis* protein sets, but with homeologs and highly related entries combined into EggNOG vertebrate-level orthology groups (*Huerta-Cepas et al., 2016*), based on the method developed in *McWhite et al., 2020*. In order to identify proteins statistically significantly associated with each bait, we calculated both a log$_2$ fold-change and a Z-score for each protein based on the observed PSMs in the bait ('expt') versus control ('ctrl') pulldown. The fold-change was computed for each protein as:

$$FC_{protein\,i} = log_2\left(\frac{(PSM_{i,expt}+1)/\sum_{j=1}^{n}(PSM_{j,expt}+1)}{(PSM_{i,ctrl}+1)/\sum_{k=1}^{n}(PSM_{k,ctrl}+1)}\right),$$

where *n* is the total number of proteins considered in the experiment.

For visualization purposes and initial rankings, we calculated significance for protein enrichment in the experiment relative to control using a one-sided Z-test as in *Lu et al., 2007* with a 95% confidence threshold (z ≥ 1.645), as:

$$Z_{protein\,i} = \frac{f_{i,expt} - f_{i,ctrl}}{\sqrt{\frac{f_{i,comb}(1-f_{i,comb})}{\sum_{j=1}^{n} PSM_{j,expt}} + \frac{f_{i,comb}(1-f_{i,comb})}{\sum_{j=1}^{n} PSM_{j,ctrl}}}},$$

e for each protein based on the where $f_i = PSM_i / \sum_{1}^{n} PSM_j$ and

$$f_{i,comb} = \left(PSM_{i,expt} + PSM_{i,ctrl}\right) / \sum_{j=1}^{n} \left(PSM_{j,expt} + PSM_{j,ctrl}\right).$$

We determined significance by calculating p-values for each Z-score using the pnorm distribution function available in the R Stats Package (v3.6.1). We corrected for multiple comparisons by computing the Benjamini-Hochberg false discovery rate using the p.adjust function, also from the R Stats Package (v3.6.1). Probability values and false discovery rates are provided in *Supplementary files 1–3*.

## Imaging, FRAP and image analysis

*Xenopus* embryos were mounted between cover glass and submerged in 1/3x MMR at stage 22 or 23, and then were imaged immediately. Live images were captured with a Zeiss LSM700 laser scanning confocal microscope using a plan-apochromat 63 × 1.4 NA oil objective lens (Zeiss) or with Nikon eclipse Ti confocal microscope with a 63×/1.4 oil immersion objective. For FRAP experiments, a region of interest (ROI) was defined as a 1.75 μm X 1.75 μm box. ROIs were bleached using 50% laser power of a 488 nm laser and a 0.64 μsec pixel dwell time. Fluorescence recovery was recorded at ~ 0.20 s intervals for up to 300 frames. Bleach correction and FRAP curve-fitting was carried out using a custom python script (modified from http://imagej.net/Analyze_FRAP_movies_with_a_Jython_script). For colocalization analysis, z- stack images were captured from at least 15 cells from at least five different embryos. Z stacks were split to single sections and ROI for DynAPs was defined in red channel (mCherry) of each section using ImageJ selection tool. The analysis was carried out using Fiji coloc2 plugin.

Intensity of dynein subunits and length of axonemes were measured using Fiji. Plots were generated using PRISM eight and the ggplot2 package in R. One-way ANOVA and Tukey's Honest Significant Difference (HSD) test and Welch Two Sample t-test were performed in R.

## Protein domain prediction

The human DAAP1 sequence was analyzed using the PSIPRED Protein Analysis Workbench for disorder and secondary structure prediction (*Buchan and Jones, 2019*).

## Airway epithelial cell culture and immunostaining

Human trachea were isolated from surgical excess of tracheobronchial segments of lungs donated for transplantation. These unidentified tissues are exempt from regulation by HHS regulation 45 CFR Part 46. Paraffin embedded tracheal sections were fixed and immunostained as previously described (*Pan et al., 2007*; *You et al., 2002*). Nuclei were stained using 4', 6-diamidino-2-phenylindole (DAPI, Vector Laboratories, Burlingame, CA, USA). The ODA protein, DNAI1, was detected using primary antibodies obtained from NeuroMab (UC Davis, Ca, clone UNC 65.56.18.11). Antibodies to DAAP1 (C16orf71) were obtained from Sigma-Millipore (St. Louis, MO, HPA049468). Images were acquired using a Zeiss LSM700 laser scanning confocal microscope.

## In situ hybridization

In situ hybridization was performed as described previously (*Sive et al., 2000*) using DIG-labeled single-stranded RNA probes against a partial sequence (1–1572) of *Xenopus c16orf71* ORF. Bright field images were captured with a Zeiss Axio Zoom. V16 stereo microscope with Carl Zeiss Axiocam HRc color microscope camera.

## Rt-pcr

To verify the efficiency of *C16orf71* MOs, MOs were injected into all cells at the 4 cell stage and total RNA was isolated using the TRIZOL reagent (Invitrogen) at stage 23. cDNA was synthesized using M-MLV Reverse Transcriptase (Invitrogen) and random hexamers. C16orf71 cDNAs were amplified by Taq polymerase (NEB) with these primers: *c16orf71* 5F 5'-cttcagagcaggacggattt-3', *c16orf71* 582R 5'-ggcagaggtgcttagatgtt-3', *c16orf71* 600F 5'- gggcttgtcattgcagtttc-3', *c16orf71* 1254R 5'-tctctaccgtccttctcttctc-3'. odc1 primers were *odc1* 426F 5'-ggcaaggaatcacccgaatg-3' and *odc1* 843R 5'-ggcaacatagtatctcccaggctc-3'.

## Data deposition

Proteomics data has been deposited into Massive which in turn was passed to ProteomeXchange.

The Massive accession # is: MSV000085075 The direct link is https://massive.ucsd.edu/ProteoSAFe/dataset.jsp?task=f6a5e10c6e114b36b8c895664860db7e.

The ProteomeXchange # is PXD017980 as noted in the paper. The direct link is http://proteomecentral.proteomexchange.org/cgi/GetDataset?ID=PXD017980.

The direct link to the data ftp site is ftp://massive.ucsd.edu/MSV000085075/.

## Acknowledgements

We thank Claire McWhite and Anna Battenhouse for consulting and advice on *X. laevis* orthogroup calculations. This work was supported by grants from the NIH R01 HL117164 and R01 HD085901 to JBW; R01 DK110520, R35 GM122480 to EMM; R01 HL128370 and R01 HL146601 to SLB; NIH K08HL150223 to AH and K99 HD092613 and LRP to KD, as well as the Welch Foundation (F-1515) to EMM. Mass spectrometry data collection was supported by CPRIT grant RP110782 to Maria Person and by Army Research Office grant W911NF-12-1-0390.

## Additional information

### Funding

| Funder | Grant reference number | Author |
|---|---|---|
| NIH | HD085901 | John B Wallingford |
| NIH | HL117164 | John B Wallingford |
| NIH | R01 DK110520 | Edward M Marcotte |
| NIH | R35 GM122480 | Edward M Marcotte |
| NIH | R01 HL128370 | Steven L Brody |
| NIH | R01 HL146601 | Steven L Brody |
| NIH | K08HL150223 | Amjad Horani |
| NIH | K99 HD092613 | Kevin Drew |
| NIH | LRP | Kevin Drew |
| Welch Foundation | F-1515 | Edward M Marcotte |
| CPRIT | RP110782 | Edward M Marcotte |
| Army Research Office | W911NF-12-1-0390 | Edward M Marcotte |

The funders had no role in study design, data collection and interpretation, or the decision to submit the work for publication.

### Author contributions

Chanjae Lee, Conceptualization, Data curation, Formal analysis, Supervision, Validation, Investigation, Visualization, Methodology, Writing - review and editing; Rachael M Cox, Data curation, Software, Methodology; Ophelia Papoulas, Data curation, Investigation, Methodology, Writing - review and editing; Amjad Horani, Formal analysis, Funding acquisition, Investigation, Methodology; Kevin

Drew, Data curation, Software, Methodology, Project administration; Caitlin C Devitt, Investigation, Methodology; Steven L Brody, Funding acquisition, Investigation, Methodology; Edward M Marcotte, Conceptualization, Data curation, Software, Formal analysis, Supervision, Funding acquisition, Methodology, Project administration, Writing - review and editing; John B Wallingford, Conceptualization, Data curation, Supervision, Funding acquisition, Visualization, Writing - original draft, Project administration, Writing - review and editing

### Author ORCIDs
Amjad Horani http://orcid.org/0000-0002-5352-1948
Edward M Marcotte https://orcid.org/0000-0001-8808-180X
John B Wallingford https://orcid.org/0000-0002-6280-8625

### Ethics
Animal experimentation: All experiments were performed in strict accordance with the UT IACU protocol # AUP-2018-00225.

### Decision letter and Author response
Decision letter https://doi.org/10.7554/eLife.58662.sa1
Author response https://doi.org/10.7554/eLife.58662.sa2

## Additional files

### Supplementary files
• Supplementary file 1. Table showing orthogroups and proteins with PSMs identified by APMS with Dnai2.

• Supplementary file 2. Table showing orthogroups and proteins with PSMs identified by APMS with Wdr78.

• Supplementary file 3. Table showing orthogroups and proteins with PSMs identified by APMS with Dnai2.

• Transparent reporting form

### Data availability
Proteomics data has been deposited into Massive which in turn was passed to ProteomeXchange. The Massive accession # is: MSV000085075 The ProteomeXchange # is PXD017980 as noted in the paper. The direct link is http://proteomecentral.proteomexchange.org/cgi/GetDataset?ID= PXD017980 The direct link to the data ftp site is ftp://massive.ucsd.edu/MSV000085075/. These data are also provided in Supp. Tables 1-3.

The following datasets were generated:

| Author(s) | Year | Dataset title | Dataset URL | Database and Identifier |
|---|---|---|---|---|
| Marcotte E, Ophelia P | 2020 | Functional partitioning of a liquid-like organelle during assembly of axonemal dyneins | https://massive.ucsd.edu/ProteoSAFe/dataset.jsp?accession=MSV000085075 | MassIVE, 10.25345/C5T69F |
| Marcotte E, Ophelia P | 2020 | Functional partitioning of a liquid-like organelle during assembly of axonemal dyneins | http://proteomecentral.proteomexchange.org/cgi/GetDataset?ID=PXD017980 | ProteomeXchange, PXD017980 |

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
