## [Decision Letter]

**Acceptance summary:**

Your study elegantly shows that the assembly of outer arm dyneins (ODAs) and inner arm dyneins (IDAs) is spatially separated in distinct cytoplasmic foci. The other important finding in this work is the identification of novel interactors of ODA and IDA subunits within these foci. In particular you identify and characterize Daap1, which is rich in intrinsically disordered regions, and show that its absence results in loss of ODAs from cilia.

**Decision letter after peer review:**

Thank you for submitting your article "Functional partitioning of a liquid-like organelle during assembly of axonemal dyneins" for consideration by *eLife*. Your article has been reviewed by three peer reviewers, and the evaluation has been overseen by a Reviewing Editor and Anna Akhmanova as the Senior Editor. The following individuals involved in review of your submission have agreed to reveal their identity: Tim Stearns (Reviewer #1); Masahide Kikkawa (Reviewer #3).

The reviewers have discussed the reviews with one another and the Reviewing Editor has drafted this decision to help you prepare a revised submission.

Summary:

Lee et al. examine liquid-like organelle "DynAPs" (axonemal dynein particles) that are important for assembly of axonemal dyneins in the cytoplasm using live-imaging and in vivo proteomics. They find DynAPs for outer arm dyneins (ODAs) and those for inner arm dyneins (IDAs) are spatially separated. The authors focused on five axonemal dynein assembly factors, three chaperones, and three dynein subunits and analyzed the co-localization of these factors in *Xenopus* MCCs (multi-ciliated cells). Detailed confocal imaging demonstrates that DynAPs are structurally partitioned for ODA and IDA subunits, which reveals discrete processing of ODA and IDA in the cytoplasm.

The other major finding in this work is the identification of novel interactors of ODA and IDA subunits in DynAPs. The authors performed affinity-purification and mass-spectrometry of Dnai2 (ODA subunit) and Wdr78 (IDA-f subunit) using *Xenopus* MCCs, and generated the sets of axonemal dynein interaction partners. Among them, Nme9, Wdr18, and Sf3a3 are analyzed and their enrichment in DynAPs are revealed. The function of an uncharacterized protein, Daap1, is analyzed and it is shown to be a novel cytoplasmic ODA regulator; Daap1 was enriched only in the ODA sub-region of DynAPs and knockdown of Daap1 caused a severe loss of ODAs from motile cilia.

Essential revisions:

All reviewers were strongly supportive of this manuscript. They agreed that it needs some extra and careful quantification, an expansion of methods and some textual edits. The key modification they suggest is quantification of data showing the subcompartmentalization of the DynAPs (Essential Revision 1). There are some suggestions below that may require new-experimental data. While these would strengthen the manuscript the reviewers would be happy to see those at a later date as a Research Advance. Their priority for now was quantification of the datasets that already exist, together with textual clarifications.

1) In Figure 2C, ODA subunit (Dnai2)-positive signals are not overlapping with IDA subunit signal (Wdr78), suggesting that DynAPs for ODA and DynAPs for IDA are mutually exclusive. Since this observation is one of the major findings in this paper, it should be shown with statistical data, not just by a few examples.

1a) Currently, only en face optical section is shown, but how do 3D stack images look like? Did the authors observe DynAP for ODA and DynAP for IDA aligned along z-direction?

1b) Statistical analyses of independent cells and DynAPs are required. Although Pearson's correlations are shown in Figure 1 and Figure 1—figure supplement 1, no similar analysis Figure 2 is shown. Without statistical analyses, one can argue that the images shown in Figure 2 can be just a few biased examples.

1c) One reviewer suggested quantification of the DynAP subcompartmentalization could be achieved by measuring the correlation of signals and suggested the following review on this topic: https://www.ncbi.nlm.nih.gov/pmc/articles/PMC3074624/.

Also: https://jcs.biologists.org/content/joces/131/3/jcs211847.full.pdf

2) Heterogeneity between DynAPs: In their previous paper, knockdown of one DNAAF (Heatr2) only reduced incidence of DynAPs to ~25% suggesting heterogeneity of DynAps could exist. If each cell has ~20 DynAPs, how many are ODA+IDA+DNAAF+ (albeit in separate subdomains), versus just DNAAF+? Just IDA+DNAAF+ versus just ODA+DNAAF+? Does this vary within the cell (a range of DynAP makeups) versus between cells (all DynAPs the same), does this change with developmental time? At the moment, it is not clear how the analysis for DynAPs was done- how each object was selected, what means were used to overcome bias. More details necessary and data could be presented as supergraphs (quantification from each cell analysed color coded) and legends (how many DynAPs analysed, from how any cells/organisms). Also revised analysis should be highly detailed in how it was performed- i.e. all DNAAF+ foci above X units in thresholding were marked in the cell and subjected to X script for correlation in FiJi in all three channels. Then vice versa with the dynein subunits. Moreover without HCs for dyneins, they can't look at heterogeneity of ODAs, but they can do different IDAs- WDR78 and DNALI1- do they colocalize in foci? This is very important to tell us whether there are there subdomains for different types of IDAs?

3) Statistics and proteomics- specificity, robustness and noise. The IP traps appear to have been run as a single run which precludes the use of t-test between replicates. Even limiting amounts of input could have been split in three for technical replicates. Although only looking at three bait-prey IP sets (not a massive dataset), instead Z-scores are chosen- where there are no replicates. There is no false discovery rate correction. More so 'hits' seem to be less specific than what is permitted by their cut-offs for significant in terms of Z-score (one-sided Z test), sometimes with only a few peptides as level. The way the data is presented is misleading for example, DNAI2 in WDR78 IP was 0.78 Z- score so not enriched significantly…. although WDR78 present in DNAI2 IP.

4) Characterization of daap1 depletion: All of interaction studies above pull out novel Daap1 which they go onto define as an ODA sub-DynAP localised protein required specifically for the correct assembly of ODA (Figure 9). The authors demonstrate this using imaging of fluorescent proteins fused to Dnai2 or Wdr78 in the axoneme only. This needs to be quantified- intensity plots against length of axoneme, next to the panels they have.

5) Control for morpholino and rescue: The authors acknowledge two separate morpholinos were used to knockdown daap1 and show depletion by RT-PCR to demonstrate effectiveness and specificity Figure 9—figure supplement 1. The norm is still to use a rescue RNA for these experiments not only to show specificity but importantly by using either the full length versus just the N-terminal DUF4701 domain of *Xenopus* Daap1 N terminal domain, you can show that the human protein is likely to have a similar effect in DynAP assembly. Or whether it's the NDK domain that is important for the function of Daap1 rather than the domain found in mammalian cells. It would strengthen analysis in Figures 7 and 8 in terms of narrative. Another alternative would be to use he more elegant experiment would have been to over-express the human variant in the *Xenopus* morphant- can it rescue the phenotype.

---

## [Author Response]

Essential revisions:All reviewers were strongly supportive of this manuscript. They agreed that it needs some extra and careful quantification, an expansion of methods and some textual edits. The key modification they suggest is quantification of data showing the subcompartmentalization of the DynAPs (Essential Revision 1). There are some suggestions below that may require new-experimental data. While these would strengthen the manuscript the reviewers would be happy to see those at a later date as a Research Advance. Their priority for now was quantification of the datasets that already exist, together with textual clarifications.

We are grateful that the reviewers consider the paper nearly complete, and we have addressed all the essential revisions by adding new quantification or new data, as appropriate. As the indicated by the Editor, we defer some suggestions to a possible Research Advance in the future.

1) In Figure 2C, ODA subunit (Dnai2)-positive signals are not overlapping with IDA subunit signal (Wdr78), suggesting that DynAPs for ODA and DynAPs for IDA are mutually exclusive. Since this observation is one of the major findings in this paper, it should be shown with statistical data, not just by a few examples.1a) Currently, only en face optical section is shown, but how do 3D stack images look like? Did the authors observe DynAP for ODA and DynAP for IDA aligned along z-direction?1b) Statistical analyses of independent cells and DynAPs are required. Although Pearson's correlations are shown in Figure 1 and Figure 1—figure supplement 1, no similar analysis Figure 2 is shown. Without statistical analyses, one can argue that the images shown in Figure 2 can be just a few biased examples.1c) One reviewer suggested quantification of the DynAP subcompartmentalization could be achieved by measuring the correlation of signals and suggested the following review on this topic: https://www.ncbi.nlm.nih.gov/pmc/articles/PMC3074624/.Also: https://jcs.biologists.org/content/joces/131/3/jcs211847.full.pdf

As requested, we have now quantified all results in Figure 2. As shown in the new Figure 2—figure supplement 1A, our analysis supports the original conclusion: Two ODA proteins, Dnai1 and Dnal4, show very strong correlation with Dnai2, while two distinct IDA proteins, Wdr78 and Dnali1, each display significantly reduced correlation with Dnai2.

2) Heterogeneity between DynAPs: In their previous paper, knockdown of one DNAAF (Heatr2) only reduced incidence of DynAPs to ~25% suggesting heterogeneity of DynAps could exist. If each cell has ~20 DynAPs, how many are ODA+IDA+DNAAF+ (albeit in separate subdomains), versus just DNAAF+? Just IDA+DNAAF+ versus just ODA+DNAAF+? Does this vary within the cell (a range of DynAP makeups) versus between cells (all DynAPs the same), does this change with developmental time? At the moment, it is not clear how the analysis for DynAPs was done- how each object was selected, what means were used to overcome bias. More details necessary and data could be presented as supergraphs (quantification from each cell analysed color coded) and legends (how many DynAPs analysed, from how any cells/organisms). Also revised analysis should be highly detailed in how it was performed- i.e. all DNAAF+ foci above X units in thresholding were marked in the cell and subjected to X script for correlation in FiJi in all three channels. Then vice versa with the dynein subunits. Moreover without HCs for dyneins, they can't look at heterogeneity of ODAs, but they can do different IDAs- WDR78 and DNALI1- do they colocalize in foci? This is very important to tell us whether there are there subdomains for different types of IDAs?

We agree this is an important point, and we have added new data to the paper that represents an initial effort to address the issue.

The revised Figure 2 shows the co-localization of sub-units from two different IDAs. We compared the IDA-f protein Wdr78 to the IDA-a,b,c protein Dnali1, finding that while the two were consistently localized adjacent one another, the actual co-localization was weak. By contrast, when we compare two IDA-f proteins (Wdr78 and Tctex1d2), they were very strongly co-localized. We quantified these results by Pearson correlation in Figure 2—figure supplement 1B, and we provide careful details of our quantification.

We thank the reviewer for suggesting this analysis, and the revised manuscript now discusses this finding, stating that IDA-subtypes are further partitioned in the structures we termed “IDA sub-DynAPs.”

We note however, that this is a very complex issue, so a fuller exploration would not fall within the scope of the present work. As the Editor suggests, we hope that further exploration of the issue can wait for a “Research Advance” in the future.

3) Statistics and proteomics- specificity, robustness and noise. The IP traps appear to have been run as a single run which precludes the use of t-test between replicates. Even limiting amounts of input could have been split in three for technical replicates. Although only looking at three bait-prey IP sets (not a massive dataset), instead Z-scores are chosen- where there are no replicates. There is no false discovery rate correction. More so 'hits' seem to be less specific than what is permitted by their cut-offs for significant in terms of Z-score (one-sided Z test), sometimes with only a few peptides as level. The way the data is presented is misleading for example, DNAI2 in WDR78 IP was 0.78 Z- score so not enriched significantly…. although WDR78 present in DNAI2 IP.

This is also an important point. To address the issue directly, we performed technical replicates on all three of our APMS experiments using reserved protein lysate from the original samples. For each of the three baits, we now present the data from both technical replicates, as well as a plot illustrating reproducibility between the two (revised Figure 3—figure supplement 1).

Importantly, these replicates helped to resolve cases with very small numbers of observed peptides. This more-thorough analysis retained all but one of the positive controls mentioned in the original manuscript (which has been removed from the revised manuscript. Moreover, all of the novel interactors described in this paper (Nme9, Wdr16, C16orf71, Sf3a3) became higher confidence predictions after the replicate.

On the other hand, as the reviewer notes, the potential Wdr78/Dnai2 interaction was less well supported after the replicates were examined. This assertion has been removed from the paper, as suggested.

4) Characterization of daap1 depletion: All of interaction studies above pull out novel Daap1 which they go onto define as an ODA sub-DynAP localised protein required specifically for the correct assembly of ODA (Figure 9). The authors demonstrate this using imaging of fluorescent proteins fused to Dnai2 or Wdr78 in the axoneme only. This needs to be quantified- intensity plots against length of axoneme, next to the panels they have.

These phenotypes have now been quantified using several metrics.

First, we generated intensity plots along axonemes that could be clearly distinguished and quantified the mean pixel intensity of ODA and IDA reporters normalized against membrane-RFP. This unbiased metric revealed highly significant reductions in ODA intensity, and significant, but much less significant, reduction in mean IDA intensity.

Second, as noted in the original manuscript, we believe the reduction in IDA intensity can be accounted for by loss specifically of the *distal* IDAs. We therefore quantified the distance from the axoneme tip as reported by membrane-RFP to the onset of signal for IDA-GFP to quantify this loss of distal IDAs.

5) Control for morpholino and rescue: The authors acknowledge two separate morpholinos were used to knockdown daap1 and show depletion by RT-PCR to demonstrate effectiveness and specificity Figure 9—figure supplement 1. The norm is still to use a rescue RNA for these experiments not only to show specificity but importantly by using either the full length versus just the N-terminal DUF4701 domain of *Xenopus* Daap1 N terminal domain, you can show that the human protein is likely to have a similar effect in DynAP assembly. Or whether it's the NDK domain that is important for the function of Daap1 rather than the domain found in mammalian cells. It would strengthen analysis in Figures 7 and 8 in terms of narrative. Another alternative would be to use he more elegant experiment would have been to over-express the human variant in the *Xenopus* morphant- can it rescue the phenotype.

We regret that we were not able to rescue the morphant phenotype with full-length mRNAs, but we hasten to add that this is not terribly uncommon in *Xenopus*, especially for late stage phenotypes. Critically, we used two very different MOs that target distinct regions of the Daap1 transcript. Moreover, we achieve a highly specific phenotype: Ciliogenesis is normal, but beating is defective, and ODAs are very preferentially affected over IDAs. We hope the reviewers will be convinced that this experiment is adequately controlled.